# Impact of Implementing Antenatal Syphilis Point-of-Care Testing on Maternal Mortality in KwaZulu-Natal, South Africa: An Interrupted Time Series Analysis

**DOI:** 10.3390/diagnostics9040218

**Published:** 2019-12-10

**Authors:** Tivani P. Mashamba-Thompson, Paul K. Drain, Desmond Kuupiel, Benn Sartorius

**Affiliations:** 1Discipline of Public Health Medicine, School of Nursing and Public Health, University of KwaZulu-Natal, Durban 4041, South Africa; Mashamba-Thompson@ukzn.ac.za (T.P.M.-T.); benn.sartorius1@lshtm.ac.uk (B.S.); 2International Clinical Research Center, Department of Global Health, University of Washington, Seattle, WA 98195-7965, USA; pkdrain@uw.edu; 3Division of Infectious Diseases, Department of Medicine, University of Washington, Seattle, WA 98195-7965, USA; 4Department of Epidemiology, University of Washington, Seattle, WA 98195-7965, USA; 5Department of Surgery, Harvard University, Massachusetts General Hospital, Boston, MA 02114, USA; 6Faculty of Infectious and Tropical Diseases, London School of Hygiene and Tropical Medicine, London WC1H 9SH, UK

**Keywords:** syphilis, point-of-care testing, maternal mortality, interrupted time series, segmented regression analysis

## Abstract

Background: Syphilis infection has been associated with an increased risk of HIV infection during pregnancy which poses greater risk for maternal mortality, and antenatal syphilis point-of-care (POC) testing has been introduced to improve maternal and child health outcomes. There is limited evidence on the impact of syphilis POC testing on maternal outcomes in high HIV prevalent settings. We used syphilis POC testing as a model to evaluate the impact of POC diagnostics on the improvement of maternal mortality in KwaZulu-Natal, South Africa. Methods: We extracted 132 monthly data points on the number of maternal deaths in facilities and number of live births in facilities for 12 tertiary healthcare facilities in KwaZulu-Natal (KZN), South Africa from 2004 to 2014 from District Health Information System (DHIS) health facility archived. We employed segmented Poisson regression analysis of interrupted time series to assess the impact of the exposure on maternal mortality ratio (MMR) before and after the implementation of antenatal syphilis POC testing. We processed and analyzed data using Stata Statistical Software: Release 13. (Stata, Corp LP, College Station, TX, USA). Results: The provincial average annual maternal mortality ratio (MMR) was estimated at 176.09 ± 43.92 ranging from a minimum of 68.48 to maximum of 225.49 per 100,000 live births. The data comprised 36 temporal points before the introduction of syphilis POC test exposure and 84 after the introduction in primary health care clinics in KZN. The average annual MMR for KZN from 2004 to 2014 was estimated at 176.09 ± 43.92. A decrease in MMR level was observed during 2008 after syphilis POC test implementation, followed by a rise during 2009. Analysis of the MMR trend estimates a significant 1.5% increase in MMR trends during the period before implementation and 1.3% increase after implementation of syphilis POC testing (*p* < 0.001). Conclusion: Although our finding suggests a brief reduction in the MMR trend after the implementation of antenatal syphilis POC testing, a continued increase in syphilis rates is seen in KwaZulu-Natal, South Africa. The study used one of the most powerful quasi-experimental research methods, segmented Poisson regression analysis of interrupted time series to model the impact of syphilis POC on maternal outcome. The study finding requires confirmation by use of more rigorous primary study design.

## 1. Background

South Africa accounts for 18% of global HIV infections, with approximately 6.7 million people infected [1]. There are almost 1000 new HIV infections per day, the majority of which are heterosexually transmitted [2]. The 2019 Statistics South Africa National Estimates report shows an ongoing increase in total number of persons living with HIV in South Africa [3]. This number has increased from an estimated 4.64 million in 2002 to 7.97 million by 2019 [3]. The report also shows that over a fifth of South African women in their reproductive ages (15–49 years) are HIV positive [3]. A recent report on maternal health indicates that non-pregnancy-related infections due to HIV/AIDS contribute to 43.7% of the total maternal mortality in South Africa [4]. The 2014 Amnesty International report on maternal health in KwaZulu-Natal (KZN) and Mpumalanga 2008–2010 highlights the barriers, such as poor access to health care, resulting in women and girls delaying or avoiding antenatal care in and maternal deaths [5].

Point-of-care (POC) testing is one innovation that has been proven to help improve healthcare access by enabling early diagnosis and linkage to care [6]. Syphilis point-of-care (POC) testing is provided as part of the routine antenatal services in South Africa. This was implemented to help improve healthcare access and neonatal health outcomes [7,8,9,10,11]. The introduction of syphilis POC testing has been shown to be effective in strengthening health systems by improving access to quality-assured prenatal screening and saving new-born lives in resource-limited settings [12,13]. The results of our survey which was aimed at determining the availability and utility of POC diagnostics in rural KZN has demonstrated that syphilis testing was available and used in all districts, but the level of availability and use varied from clinic to clinic [14]. Overall 50% (CI: 60–40%) of the clinics used syphilis tests [14].

Syphilis infection has been associated with HIV through facilitating HIV transmission and the combination of syphilis, particularly during pregnancy [9,15]. Early detection of *Treponema pallidum* and prompt penicillin treatment for pregnant women who test positive have been shown to be effective in reducing adverse pregnancy outcomes [16,17,18]. Syphilis has also been demonstrated to increase HIV viral load and decrease CD4 cell counts in HIV-infected patients with syphilis infections [19,20]. Atypical presentations of early syphilis, rapid progression to tertiary syphilis, treatment failures, and more frequent cases of neurosyphilis have been reported amongst HIV-infected populations [15]. This is a concern for low- and middle-income countries, such as South Africa, that have high rates of HIV-related maternal mortality [5,21].

Despite this, little is known about the impact of syphilis POC testing on maternal mortality, particularly in high HIV settings. The results of our systematic review revealed that there is limited evidence on the impact of syphilis POC diagnostics on maternal outcomes of HIV-infected women [4]. The main aim of this study is to determine the impact of antenatal syphilis POC diagnostics on maternal mortality in KwaZulu-Natal, South Africa, using a time series study design. Demonstrating the impact of currently used syphilis POC testing on maternal mortality will enable us to determine whether or not the introduction of syphilis POC diagnostics has had any tangible effect on key maternal outcomes to enable justification of the need for syphilis POC diagnostic scale-up in settings that lack laboratory infrastructure such as rural antenatal health clinics.

## 2. Methodology

This study was conducted as part of a large implementation research study aimed at evaluating the accessibility and utility of POC diagnostics for maternal health in rural South Africa primary healthcare clinics (PHC) in order to generate a model framework of implementation of POC diagnostics in rural South African clinics [22].

### 2.1. Study Design, Population, and Location

We conducted a retrospective study using monthly MMR data from all 11 districts in KZN province, South Africa from 2004 to 2014. KZN province was purposively selected due to the high prevalence of HIV and high maternal mortality. A quasi-experimental approach is a powerful approach for evaluating effects of presence and absence of an exposure on outcomes [23,24].

### 2.2. Exposure

As reported by the KZN Department of Health, Rapid Plasma Reagin (RPR; Biotec Lab, Suffolk, UK) syphilis POC testing was implemented as part of the routine diagnostic tests for antenatal care in primary healthcare facilities during 2007. Syphilis POC test forms one the routine antenatal diagnostic tests offered by nurses at PHC clinics. Prior to the introduction of syphilis POC testing in this region, syphilis testing was performed at the laboratory associated with the health facility using Lovibond (Orbeco-Hellige, FL, USA) and Rapid Plasma Reagin (RPR; Biotec Lab, Suffolk, UK), respectively.

### 2.3. Data Extraction

We extracted archived records on the number of annual live births and number of maternal deaths in KZN from existing routine data from the District Health Information System (DHIS; Appendix A). Earlier studies of the DHIS system have reported that the quality of the data, including those used to track prevention of mother to child transmission (PMTCT) care, is suboptimal [25]. Following these reports, in 2008 the KwaZulu-Natal Department of Health, the University of KwaZulu-Natal, and the Institute for Health care Improvement launched a large-scale effort, entitled the 20,000+ Partnership, to improve the completeness and accuracy of the public health data routinely recorded in the DHIS implemented between May 2008 and March 2009 [26]. This exposure has led to improved data reliability and validity [26].

We extracted data from KZN tertiary facilities that contained data on number of maternal deaths in the facility and number of live births in the facility from 2004 to 2014. Data prior to 2004, after 2014, and other data that were not relevant to maternal health were excluded. The DHIS presents district level data on all the maternal health indicators required to calculated MMR. We extracted district level data on number of maternal deaths in facilities and number of live births in facilities to help us calculate MMR.

### 2.4. Data analysis

We processed and analyzed data using Stata Statistical Software: Release 13. (College Station, TX: StataCorp LP). The exposure (syphilis POC test) was implemented as part of the routine diagnostic tests for antenatal care in primary healthcare facilities during 2007. Hence, we conducted a pre-post analysis of this exposure. Estimation of the impact of syphilis POC diagnostics on overall MMR in all KZN districts was conducted using the exposure time series approach [27]. We used the Poisson formulation approach, given maternal death count (with live births as the exposure) to estimate changes in levels and trends in maternal deaths during the period before and after implementation of syphilis POC testing in KZN, using the following equation:Ŷ*_t_* = log *(L_t_)* + *β*_0_ + *β*_1_ × time + *β*_2_ × exposure + *β*_3_ × time after exposure + e*_t_*

Where Ŷ*_t_* is the outcome (maternal deaths); L*_t_* is the live birth count; time indicates the number of months from the start of the series; exposure is the dummy variable taking the values 0 in the pre-exposure segment and 1 in the post-exposure segment; time after exposure is 0 in the pre-exposure segment and counts the months in the post-exposure segment at time t; the coefficient *β_0_* estimates the base level of the outcome (MMR) at the beginning of the series; *β_1_* estimates the base trend, i.e., the change in outcome per month in the pre-exposure segment; *β_2_* estimates the change in level of MMR on the post-exposure segment; *β_3_* estimates the change in trend in MMR in the post-exposure segment; and e*_t_* estimates the error

We used line graphs to visually display the series over time namely, one curve represented the observed MMR by year and month while a second line displayed the predicted line from the segmented regression model. Model coefficients were exponentiated to represent relative risks (RR).

### 2.5. Ethics Statement

We received full ethical approval and permission to conduct this study from the KZN Department of Heath (DoH) Ethics Committee (HRKM 40/15). We also received ethical approval for the current study from the University of KwaZulu-Natal (UKZN) Biomedical Research Ethics Committee (BE484/14). Data used in this study was extracted from the national database. Therefore, patient informed consent was not required.

## 3. Results

### Summary of the Study Population and Sample Size

A total of 132 consecutive monthly MMR data points from 12 tertiary healthcare facilities from all 11 KZN districts were assessed from 2004 to 2014. The data comprised of 36 temporal points before the introduction of syphilis POC test exposure and 84 after the introduction in primary health care clinics (Appendix A). The average annual MMR for KZN from 2004 to 2014 was estimated at 176.09 ± 43.92 ranging from a minimum of 68.48 to a maximum of 225.49 (Table 1). Table 1 shows a drop in MMR by 34.2% in 2008 followed by a 35.5% rise in 2009.

Figure 1 shows an annual increase in maternal mortality ratio (MMR) level from 2004 to 2014, with a prominent drop in MMR level in 2008. Figure 2 shows the average and monthly fluctuations in MMR levels. A 35% decrease in MMR level was estimated during implementation of the syphilis POC test (*p* < 0.001; Table 2).

Table 2 and Figure 3 depict results from the segmented Poisson regression (interrupted time series) model. Based on the smoothed line plots and the reported relative risk (RR), there is a significant decrease in the monthly MMR before and after the syphilis POC testing exposure in KZN (35%). A significant 1.5% increase in MMR was observed (*p* < 0.001) before the introduction of syphilis POC testing exposure (2004–2006). MMR during the period after the introduction of syphilis POC testing exposure (2008–2014) rose significantly by 1.3% every month (*p* < 0.001).

## 4. Discussion

Our study shows that implementation of antenatal syphilis POC tests has the potential to improve maternal mortality in high HIV prevalence regions. A higher increase in MMR trend during the period before implementation (2004–2006) and a relatively lower increase MMR trend after implementation of antenatal syphilis POC test exposure (2008–2014) was demonstrated. The increase in MMR trends before and after the exposure were significant.

To the best of our knowledge, this is the first study that has evaluated the impact of syphilis POC testing on MMR in a high HIV prevalence resource-limited setting. Our recent systematic review has shown a lack of evidence on the impact of syphilis POC diagnostics on maternal outcomes of HIV-infected patients [4]. A quasi-experimental interrupted time series design was employed to determine the impact of syphilis POC testing on maternal mortality ratio (MMR) in KZN. An interrupted time series is the most powerful quasi-experimental approach for evaluating effects of presence and absence of an exposure on outcomes [24,28]. This study design has enabled estimation of the change in MMR levels during the period of implementation, before and after implementation of syphilis POC test intervention, leading to determination of potential exposure effect on maternal mortality. This sharp change in the MMR trend marked the beginning of the second segment of the interrupted time series and does not reflect the immediate effect of discontinuation of syphilis POC testing intervention. Therefore, the interruption will occur at this data point (2007). The trend changes after this data point (2007) reflect the potential effect of introduction of syphilis POC testing in KZN. It is also worth noting the sharp drop after 2007 and a sharp rise in MMR in 2009, which was caused by missing data during 2008. The model used this data analysis included an omission command for the 2008 data points. Therefore the 2008 missing data was disregarding during analysis to ensure reliability of the model output.

DHIS data quality was a major limitation in this study [25]. Moreover, although syphilis infections have been associated with an increased risk of HIV transmission among pregnant women, the prevalence of syphilis is low in KZN. Therefore, in comparison to other disease conditions such as HIV, which have a high prevalence among women in KZN [29], the introduction of syphilis POC testing as a single exposure will not have a substantial impact on maternal mortality. However, due to the POC implementation dates (1998) for HIV and syphilis (2007) and the lack of archived DHIS MMR records prior to 2004, syphilis POC testing was the most suitable exposure for this study. During the analysis of monthly the MMR, data outliers which may have shifted the slope of the regression segments were detected. The sudden drop (34.2%) in MMR from 2007 to 2008 demonstrates inconsistencies in the data set. This finding is supported by a previous study which was aimed at introducing data improvement exposure for indicators in selected clinics in KZN in May 2008 [26]. This study revealed a significant improvement in data completeness from 26% before introduction of the exposure to 64% after the exposure, and also demonstrated a significant increase in data accuracy from 37% to 65% (*p* < 0.0001) after the exposure [26]. This was a retrospective study that used available routine DHIS data to answer the research question. The DHIS data did not contain the number of patients who were exposed to syphilis testing during each year. However, we are aware that all pregnant women are offered syphilis tests as part of routine antenatal test. Another limitation of this study is that our model only used DHIS data, which did not include data on pre-eclampsia cases. Pre-eclampsia is the leading cause of maternal and fetal morbidity and mortality worldwide [30,31,32]. It is a pregnancy-induced disorder reported to cause complications in approximately 5–7% of pregnancies.

The 2011 World Health Organization (WHO) global HIV/AIDS response has shown that up to one-third of the women attending antenatal care clinics are not tested for syphilis [33]. Maternal syphilis infections continue to affect large numbers of pregnant women, causing substantial perinatal morbidity and mortality that could be prevented by early testing and treatment [34,35]. The use of syphilis POC testing has been shown to result in health system strengthening and saving newborn lives [12]. Bearing in mind the reported increase in HIV-related maternal mortality in KZN [36] and the potential impact of syphilis POC testing on reducing MMR demonstrated in this study, there is a need for scaling up of syphilis POC testing. Although the results show the potential impact of syphilis POC testing on reducing maternal mortality, the level of MMR is still increasing. This demonstrates the need for additional interventions to deduce maternal mortality post-intervention.

As POC tests are being increasingly designed for use in resource-limited settings [37,38], rigorous assessment of the impact of current and future POC tests on key health outcomes is crucial in order to justify scale up or test replacement. This is supported by De Schacht et al. studies which recommend the need to identify optimal health delivery strategies to effectively bring the impact of technological advances such as POC testing to patients that are most at need [13]. Due to the recently reported increasing prevalence of HIV among women in KZN [29], detection of treatable infections that are associated with HIV transmission is crucial for the reduction of HIV-related maternal mortality. Future modelling studies on the impact of POC tests in KZN should exclude MMR data collected in 2008 to improve the reliability of the results.

## 5. Conclusions

The results of this study show an increasing maternal mortality ratio in KZN, South Africa. It also demonstrated the potential impact of antenatal syphilis POC test exposure on reducing the increase in MMR. However, the impact was in consideration of confounding factors related to MMR in these settings, thus the results require confirmation by use of a more rigorous study design. Bearing in mind the high importance of improving maternal outcomes in high HIV prevalent regions, efforts to help in the continual improvement of POC testing interventions aimed at improving maternal mortality are essential, particularly in these settings.

## Figures and Tables

**Figure 1 diagnostics-09-00218-f001:**
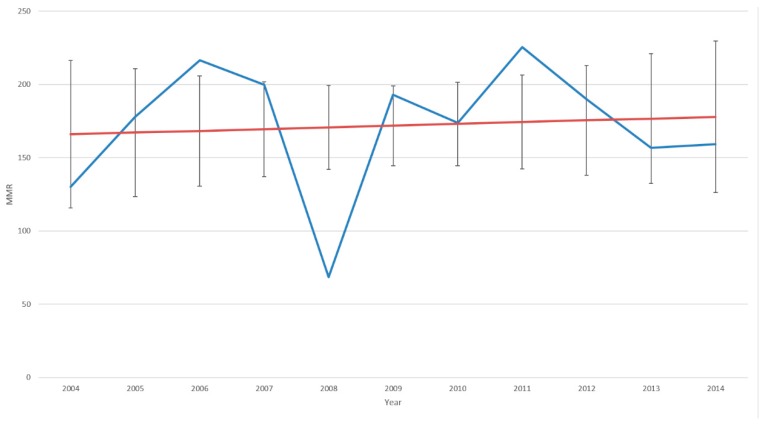
Maternal mortality ratio (MMR) by year for all KwaZulu-Natal with trend line (and 95% CI) excluding 2005 outlier (detected from raw data).

**Figure 2 diagnostics-09-00218-f002:**
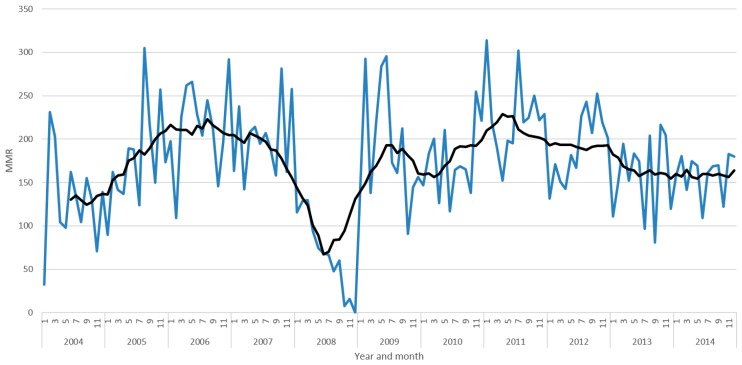
Maternal mortality ratio (MMR) by year for all KZN by month with 12 months moving average line (black).

**Figure 3 diagnostics-09-00218-f003:**
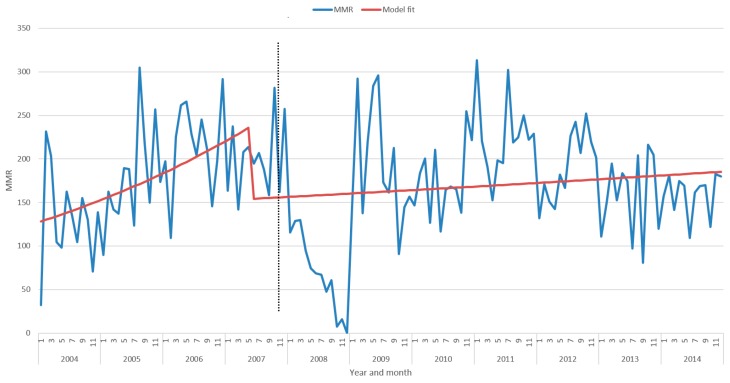
MMR by year for all KZN by month with fitted interrupted time series segmented regression line (exposure implementation represented by vertical dotted line).

**Table 1 diagnostics-09-00218-t001:** Maternal mortality rate by year for all KwaZulu-Natal.

Year	Number of Maternal Deaths in Facility	Number of Live Births in Facility	Maternal Mortality Ratio Per 100,000 Live Births
2004	208	159712	130.23
2005	259	145617	177.86
2006	371	171230	216.67
2007	337	168580	199.91
2008	120	175227	68.48
2009	324	167847	193.03
2010	291	167544	173.69
2011	378	167637	225.49
2012	312	164322	189.87
2013	256	163411	156.66
2014	254	159484	159.26

**Table 2 diagnostics-09-00218-t002:** Longitudinal analysis of monthly maternal mortality using a fully segmented Poisson regression model.

Full Segmented Regression Model	RR	95% CI		*p* Value
		Lower	Upper	
A_0_—Intercept i (2007, introduction of POC testing)	0.001	0.001	0.001	---
A_1_—Baseline trend (before introduction of syphilis POC testing)	1.015	1.01	1.021	<0.001
A_2_—Level change after introduction of syphilis POC testing	0.653	0.567	0.752	<0.001
A_3_—Trend change introduction of syphilis POC testing	0.987	0.981	0.992	<0.001

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
