# Peer review of "Impact of Implementing Antenatal Syphilis Point-of-Care Testing on Maternal Mortality in KwaZulu-Natal, South Africa: An Interrupted Time Series Analysis"

_diagnostics, 2019, doi:10.3390/diagnostics9040218_

Round 1

Reviewer 1 Report

Thank you for giving me the opportunity to review the article. The authors conducted a study on the impact of implementing antenatal syphilis POC testing on maternal mortality in KZN. The topic was interesting, but there were major concerns in the manuscript. Not only that, the data quality of this study is crucial limitation as the authors mentioned. Therefore, I thought that the manuscript cannot be accepted for publication in the journal Diagnostics. I left the major comments for future submission to other journals.

Major Comments:

Abstract:

P1: The value “176.09, ± 43, 92” means “176.09 ± 43.92”?

Introduction:

P2: The HIV prevalence report should be updated. It because that the current cited report published 6 years ago.

Methods:

P2: How to collect the MMR data in 11 districts in KZN? The data corrected qualified methods and the same procedure used in these districts? P2: This is the retrospective study; therefore, the authors should not use the word “intervention” (“exposure” is better to be used). P2-3: Did the authors evaluate the coverage rate of the POC test? The reviewer would like to know that most of the potential subjects provided POC test. P3: The characteristics of “archived records” (reliability and validity of the record) should be mentioned. P3: The authors should add a statement about data availability. P3: The difference between facilities should also be evaluated (as a secondary outcome).

Results:

P6: A large variation of MMR in each month was observed as shown in MMR. Therefore, it should be difficult to discuss the results based on linear regression.

Author Response

We are most grateful for your time to review this manuscript. Yours comments were useful and have help improve this manuscript significantly.

Responses to Reviewer #1 comments

Thank you for giving me the opportunity to review the article. The authors conducted a study on the impact of implementing antenatal syphilis POC testing on maternal mortality in KZN. The topic was interesting, but there were major concerns in the manuscript. Not only that, the data quality of this study is crucial limitation as the authors mentioned. Therefore, I thought that the manuscript cannot be accepted for publication in the journal Diagnostics. I left the major comments for future submission to other journals.

Major Comments:

Abstract:

Comment: P1: The value “176.09, ± 43, 92” means “176.09 ± 43.92”?

Response: We have corrected this minor error in the abstract. 176.09, ± 43, 92” has been changed to “176.09 ± 43.92. Line 28.

Introduction:

Comment: P2: The HIV prevalence report should be updated. It because that the current cited report published 6 years ago.

Response: We have updated the HIV prevalence report.  Line 48 to 51

Methods:

Comment: P2: How to collect the MMR data in 11 districts in KZN?

Response: We extracted from routine existing routine data from the District Health Information System (DHIS). The DHIS presents district level data on all the maternal health indicators required to calculated MMR. We extracted district level data on number of maternal deaths in facility and number of live births in facility to help us calculate MMR. Line 112-114

Comment: The data corrected qualified methods and the same procedure used in these districts?

Response: Yes, we can confirm that the collected data was the most appropriate to answer research question of this study.

Comment: P2: This is the retrospective study; therefore, the authors should not use the word “intervention” (“exposure” is better to be used).

Response: We have replaced “intervention” with “exposure” through the manuscript.

Comment:  P2-3: Did the authors evaluate the coverage rate of the POC test?

Response: We have conducted a survey to determine the availability and accessibility of POC tests in all 11 KZN districts. The results of the survey were published in BMC Health Service Research journal [1]. The results demonstrated that syphilis test was available and used in all districts but the level of availability and use varied from clinic to clinic. Overall 50% (CI: 0.6–0.4) of the clinics used syphilis tests [1]. The survey and the current were conducted in 2015 and the data used for the current study was from 2004 to 2014.

Comment:  The reviewer would like to know that most of the potential subjects provided POC test.

Response: Syphilis POC testing as routine test to pregnant women as part of the antenatal care package and available at primary healthcare clinics in South Africa. It is also provided to patients with symptoms of sexual transmitted diseases. It is not offered to regular patients who do not fit into the above criteria.

Comment: P3: The characteristics of “archived records” (reliability and validity of the record) should be mentioned.

Response: Earlier studies of the DHIS system have reported that the quality of the data, including those used to track PMTCT care, is suboptimal [2]. Following these reports, in 2008 the KwaZulu-Natal Department of Health, the University of KwaZulu-Natal and the Institute for Health care Improvement launched a large-scale effort, entitled the 20000+ Partnership, to improve the completeness and accuracy of the public health data routinely recorded in the DHIS implemented between May 2008 and March 2009 [3]. This intervention has led to improved data reliability and validity [3].

Comment: P3: The authors should add a statement about data availability.

Response: We added the following statement. Line 254 to 260.

Comment: P3: The difference between facilities should also be evaluated (as a secondary outcome).

Response: Assessing the difference between facilities is outside the scope of our study and not feasible with the dataset. However, were have presented the number of maternal deaths in facility and number of live births in the included tertiary healthcare facilities. Line 147 to 152.

Results:

Comment: P6: A large variation of MMR in each month was observed as shown in MMR. Therefore, it should be difficult to discuss the results based on linear regression.

Response: Thank you for your comment. We agree with the comments, yes this would be very difficult to calculate without the use of the model. We have discussed the results produced by the model.

Reviewer 2 Report

Dear Authors,

congratulations on the meticulous effort you have put into preparing this manuscript. I think it will be both interesting and valuable for readers from various backgrounds. I would like to consider introducing some minor improvements prior to publication in Diagnostics.

In paragraph 2.2 you describe POC and standard testing - could you include what percentage of population has been screened in both situations in consecutive years? Paragraph 2.4 Data analysis is to detailed - please describe only functions crucial to understand your reasoning, especially Table 2. Could you clearly explain - POC screening was used only in year 2007? How was it organized and financed? In Discussion - please elaborate on confounding factors
related to MMR in KwaZulu. Was there any change that could be the reason for increase in MMR like famine, war, infection outbreak?

Author Response

We sincerely appreciate your time used to review this manuscript. Your comments were useful and have helped improve this manuscript significantly.

Responses to Reviewer #2 comments

Comment: Congratulations on the meticulous effort you have put into preparing this manuscript. I think it will be both interesting and valuable for readers from various backgrounds. I would like to consider introducing some minor improvements prior to publication in Diagnostics.

Comment: In paragraph 2.2 you describe POC and standard testing - could you include what percentage of population has been screened in both situations in consecutive years?

Response: This was a retrospective study that used available routine DHIS data to answer the research question. The DHIS data did not contain the number of patients who were exposed to syphilis testing, we have included this in the discussion section. Line 2010-213

Comment: Paragraph 2.4 Data analysis is to detailed - please describe only functions crucial to understand your reasoning, especially Table 2. Could you clearly explain - POC screening was used only in year 2007?

Response: The exposure (syphilis POC test) was implemented as part of the routine diagnostic tests for antenatal care in primary healthcare facilities during 2007. We conducted a pre-post intervention analysis of this exposure. Line 117 to 119.

Comment: How was it organized and financed?

Response: Syphilis POC testing is provided as part of the routine antenatal services in South Africa. This is part of the government funded healthcare services provided in primary healthcare clinics. Line 95-97

Comment: In Discussion - please elaborate on confounding factors related to MMR in KwaZulu. Was there any change that could be the reason for increase in MMR like famine, war, infection outbreak?

Response: We have elaborated on confounding factors related to MMR in KwaZulu-Natal. Line 214-218

Reviewer 3 Report

I read with great interest the Manuscript titled “Impact of Implementing Antenatal Syphilis Point-of Care Testing on Maternal Mortality in KwaZulu-Natal, South Africa: An Interrupted Time Series Analysis” (diagnostics-617230), which falls within the aim of Diagnostics.    

In my honest opinion, the topic is interesting enough to attract the readers’ attention. Methodology is accurate and conclusions are supported by the data analysis. Nevertheless, authors should clarify some point and improve the discussion citing relevant and novel key articles about the topic.

Authors should consider the following recommendations:

Manuscript should be further revised by a native English speaker. Inclusion/exclusion criteria should be better clarified. The Authors did not mention the sample size calculation for their study. It is essential to specify this data in order to guarantee an adequate significance of the results obtained by the Authors. I suggest to stress, at least briefly, other potential causes of maternal mortality such as pre-eclampsia (refer to: PMID: 28282763; PMID: 28243732; PMID: 29923045). According to recent data (PMID: 25444613; PMID: 27216405) incoming migrants from developing Countries seem to have little chance to undergo pregnancy screening for syphilis, Hepatitis B virus and HIV. Considering the importance of the topic, I suggest discussing it.

Author Response

Thank you for finding time out of your busy schedule to review this manuscript. Your comments were useful and have helped improve this manuscript significantly.

Responses to Reviewer # 3 comments

I read with great interest the Manuscript titled “Impact of Implementing Antenatal Syphilis Point-of Care Testing on Maternal Mortality in KwaZulu-Natal, South Africa: An Interrupted Time Series Analysis” (diagnostics-617230), which falls within the aim of Diagnostics.

Comment: In my honest opinion, the topic is interesting enough to attract the readers’ attention. Methodology is accurate and conclusions are supported by the data analysis. Nevertheless, authors should clarify some point and improve the discussion citing relevant and novel key articles about the topic.

Authors should consider the following recommendations:

Comment: Manuscript should be further revised by a native English speaker.

Response: The manuscript was reviewed by two co-authors who are native English speakers, we have also typos or grammatical errors throughout the manuscript.

Comment: Inclusion/exclusion criteria should be better clarified.

Response: We have clarified the inclusion and exclusion criteria. Line 110 to 112

Comment: The Authors did not mention the sample size calculation for their study. It is essential to specify this data in order to guarantee an adequate significance of the results obtained by the Authors.

Response: In this study, we included all data points from all 12 tertiary facilities in KwaZulu-Natal. The data comprised 36 temporal points before introduction of and 84 after introduction of syphilis POC test intervention in primary health care clinics (Appendix 1). Line 145 to 151.

Comment: I suggest to stress, at least briefly, other potential causes of maternal mortality such as pre-eclampsia (refer to PMID: 28282763; PMID: 28243732; PMID: 29923045).

Response: Our analysis was limited to the available data. Our model included all maternal mortality indicators reported in the DHIS. Unfortunately, data on pre-eclampsia, the leading course of maternal deaths was not reported. We have noted this in the limitation section of this study. Line 211 to 218.

Comment: According to recent data (PMID: 25444613; PMID: 27216405) incoming migrants from developing countries seem to have little chance to undergo pregnancy screening for syphilis, Hepatitis B virus and HIV. Considering the importance of the topic, I suggest discussing it.

Response: Thank you for sharing the articles and for the suggestion. It is alarming to see this happening in high-income countries. Fortunately, in South Africa, antenatal care is provided for all mothers including migrant mothers at primary healthcare. The use of community health workers all helps to reach all mothers in rural and resource-limited settings.

Round 2

Reviewer 1 Report

Thank you for giving me the opportunity to review the article. The authors revised the manuscript according to the comments, but additional corrections should be done before accepting. As I mentioned previously, I only left the major comments in the first round of review. Therefore, I would like the authors to revise the manuscript as per the suggestions below.

Additional Comment based on the first round of review:

Methods:

P2: This is the retrospective study; therefore, the authors should not use the word “intervention” (“exposure” is better to be used).

Authors’ Response: We have replaced “intervention” with “exposure” through the manuscript.

Additional Comment: The authors should check the section “Data analysis”, and others to avoid the use of the word “intervention.”

P2-3: Did the authors evaluate the coverage rate of the POC test? The reviewer would like to know that most of the potential subjects provided POC test.

Authors’ Response: We have conducted a survey to determine the availability and accessibility of POC tests in all 11 KZN districts. The results of the survey were published in BMC Health Service Research journal [1]. The results demonstrated that syphilis test was available and used in all districts but the level of availability and use varied from clinic to clinic. Overall 50% (CI: 0.6–0.4) of the clinics used syphilis tests [1]. The survey and the current were conducted in 2015 and the data used for the current study was from 2004 to 2014.

Additional Comment: The authors should correct the value “50% (CI: 0.6–0.4)” to “50% (95% CI: 40-60)”, if the authors calculated the 95% confidence interval.

P3: The authors should add a statement about data availability.

Authors’ Response: We added the following statement. Line 254 to 260.

Additional Comment: If the data is publicity available as the authors mentioned, they should upload the data as a supplementary material. It is very informative for the researchers who refer the data used in this study.

Additional Comments in the second round of review.

The authors stated that “However, data quality was one of the major limitations of this study. Therefore, intervention to help improve the quality of clinical data collected from clinical settings is recommended…”, but the recommendation statement should be deleted. It because that the study design is retrospective, and the authors use data which quality is not enough. This recommendation can be provided after the well-designed study which will perform in the future.

Author Response

We have uploaded our response as part of the submission

Responses to reviewers

Reviewer #1

Comment: Thank you for giving me the opportunity to review the article. The authors revised the manuscript according to the comments, but additional corrections should be done before accepting. As I mentioned previously, I only left the major comments in the first round of review. Therefore, I would like the authors to revise the manuscript as per the suggestions below.

Additional Comment based on the first round of review:

General response

Thank you for making time out of your busy schedule to review this manuscript. We sincerely appreciate your comments and suggestions. We have accordingly addressed them as you suggested. Kindly find below a point-by-point response to your comments and suggestions.

Methods:

Comment: P2: This is the retrospective study; therefore, the authors should not use the word “intervention” (“exposure” is better to be used).

Authors’ Response: We have replaced “intervention” with “exposure” through the manuscript.

Additional Comment: The authors should check the section “Data analysis”, and others to avoid the use of the word “intervention.”

Second response: We have preplaced “intervention” with “exposure in the analysis section and throughout the manuscript.

P2-3: Did the authors evaluate the coverage rate of the POC test? The reviewer would like to know that most of the potential subjects provided POC test.

Authors’ Response: We have conducted a survey to determine the availability and accessibility of POC tests in all 11 KZN districts. The results of the survey were published in BMC Health Service Research journal [1]. The results demonstrated that syphilis test was available and used in all districts but the level of availability and use varied from clinic to clinic. Overall 50% (CI: 0.6–0.4) of the clinics used syphilis tests [1]. The survey and the current were conducted in 2015 and the data used for the current study was from 2004 to 2014.

Additional Comment: The authors should correct the value “50% (CI: 0.6–0.4)” to “50% (95% CI: 40-60)”, if the authors calculated the 95% confidence interval.

Second response: We have corrected the value “50% (CI: 0.6–0.4)” to “50% (95% CI: 40-60)” for results of our confidence interval calculations. Line 67

P3: The authors should add a statement about data availability.

Authors’ Response: We added the following statement. Line 254 to 260.

Additional Comment: If the data is publicity available as the authors mentioned, they should upload the data as a supplementary material. It is very informative for the researchers who refer the data used in this study.

Response: We have attached the data set as part of the supplementary material.

Additional Comments in the second round of review.

The authors stated that “However, data quality was one of the major limitations of this study. Therefore, intervention to help improve the quality of clinical data collected from clinical settings is recommended…”, but the recommendation statement should be deleted. It because that the study design is retrospective, and the authors use data which quality is not enough. This recommendation can be provided after the well-designed study which will perform in the future.

Response: The reviewer’s suggestion was accepted. We have deleted these two sentences “However, data quality was one of the major limitations of this study. Therefore, intervention to help improve the quality of clinical data collected from clinical settings is recommended…”

Round 3

Reviewer 1 Report

Thank you for giving me the opportunity to review the article. The authors appropriately revised the manuscript according to the comments. I thought that the manuscript can be accepted for publication in the journal.